# Planting date in South Kivu, eastern DR Congo: A real challenge for the sustainable management of *Spodoptera frugiperda* (Lepidoptera: Noctuidae) by smallholder farmers

**Marcellin Cuma Cokola**[1,2]*, **Grégoire Noël**[2], **Yannick Mugumaarhahama**[3], **Rudy Caparros Megido**[2], **Espoir B. Bisimwa**[1], **Frédéric Francis**[2]

1 Faculty of Agriculture and Environmental Sciences, Department of Crop Sciences, Université Evangélique en Afrique, Bukavu, South Kivu, DR Congo, 2 Functional and Evolutionary Entomology, Gembloux Agro-Bio Tech, Liege University, Liege, Belgium, 3 Faculty of Agriculture and Environmental Sciences, Unit of Applied Biostatistics, Université Evangélique en Afrique, Bukavu, South Kivu, DR Congo

* marcellin92cokolacuma@gmail.com, m.cokola@uliege.be

**Data Availability Statement:** All relevant data are within the manuscript and its Supporting Information files.

## Abstract

There is growing research interest in the fall armyworm (FAW) *Spodoptera frugiperda*, a polyphagous insect that is a major pest of maize crops worldwide. We investigated the relationship between planting date of maize and FAW infestation in South Kivu, eastern Democratic Republic of Congo, in two sampling seasons (September to October 2020 and February to March 2021). Five planting dates were considered for 45 fields in each season. The incidence, severity of attack and larval density of FAW were assessed at the 8-leaf stage (V8) of maize development in monoculture and intercropping systems. Planting period, classified as late or early, had a strong influence on FAW larval density, incidence and severity. The results showed that the late planting period (mainly on 30 October in season-1 and 30 March in season-2) had the highest larval density, incidence and severity of attack compared to the early planting period (15 September in season-1 and 01 Mars in season-2). During the season-1, five larval stages were found in the same field, whereas all larval stages were present in season-2, regardless of planting period. High densities of L4, L5 and L6 larvae were much more associated with late planting and incidence appeared to be highest when these larvae were present. The presence of L2 and L3 larval stages was observed in maize cropping systems intercropped with soybean and peanuts, while maize in monoculture and intercropped with cassava and beans was colonized by L4, L5 and L6 larvae. This study highlights the existence of different maize planting dates in South Kivu and demonstrates that late plantings have significant FAW infestations compared to early plantings. It provides a basis for developing climate-smart integrated pest management.

**Funding:** Université Evangélique en Afrique with funds from Pain pour Le Monde (Project A-COD-2023-0035). The funders had no role in study design, data collection and analysis, decision to publish, or preparation of the manuscript.

**Competing interests:** The authors have declared that no competing interests exist.

## Introduction

Since 2016, Africa has been invaded by the fall armyworm (FAW), *Spodoptera frugiperda* (J. E. Smith) [1]. This species from tropical and subtropical America [2,3] is a highly mobile insect pest with a wide range of host plants [4,5], preferentially cereals including maize crop [6]. Currently, only the European continent has not yet undergone an invasion of the FAW [2]. Fall armyworm is a prolific species that does not undergo a diapause [7,8] and whose adult moths can migrate from one region to another when conditions are no longer optimal [9,10]. Because of its polyphagous feeding behavior, FAW can maintain its population throughout the year by infesting other crops [5,11]. In the Americas, approximately 353 species have been identified as alternate hosts of FAW based on the literature compiled by Montezano et al. [5]. On the African continent, sorghum, cabbage, Napier grass and onion have been officially reported as alternative hosts [11,12] while in Asia, sugarcane and ginger was recorded [13,14].

The fall armyworm is known to have the ability to cause huge infestations up to 100% in maize plantations [15]. Considering the phenological stages of maize, FAW attacks start once the first leaves unfold, precisely at early whorls (VE to V6 stages) and the infestation is intense at the vegetative growth stage, usually at the late whorl (stages V7, V8 to VT) [16]. The fall armyworm has a severe impact on maize crops globally, leading to yield losses, increased costs, and risks to food security, especially in developing nations [17]. Day et al. [18] estimated losses caused by FAW in the range of 8.3 to 20.6 million tons of maize each year in the absence of effective control methods in Africa. For the case of the Democratic Republic of Congo (DRC), losses may be as high as 633,000 tons/year [18]. Recent studies in Africa by Eschen et al. [19] report average losses caused by FAW on maize crops in monetary value of 9.4 billion USD. According to Overton et al. [17], there is a positive relationship between the density/infestation rate of FAW and yield reduction in maize while Harrison et al. [20] found the opposite. According to Harrison et al. [20], a variable proportion of the FAW population present in a field will experience natural mortality, considering landscape complexity and climatic conditions, and therefore the infestation rate provides little useful information on yield reduction. Due to the extent of damage on maize leaves, most farmers in Sub-Saharan Africa use synthetic chemicals [12,21]. The use of insecticides to control FAW in maize crops is often considered ineffective due to incorrect application methods and the larvae's feeding behavior, which gives them a degree of resistance to certain active molecules [22].

Sustainable management of FAW depends on knowledge of its bioecology rather than the use of synthetic insecticides [6,23]. Sustainable management methods include agricultural practices grouped in an agroecological approach [6]; semiochemical based methods that combine the use of pheromones and cropping systems in a push-pull arrangement [24]. Intercropping offers a sustainable, low-cost and environmentally friendly approach to managing FAW [6]. By increasing biodiversity in cropping systems, intercropping helps to minimize pest damage and improve overall crop health [25]. Diversified cropping systems make it harder for FAW to locate and feed on its preferred host, reducing the overall pest pressure [26]. Intercropping supports a habitat for beneficial insects such as ladybugs, spiders, and parasitoid wasps, which prey on FAW eggs and larvae [27,28]. In plant protection, manipulation of crop planting date is one of six categories of preventive actions against crop pests [29]. For example, Slosser [30] measured the influence of planting date on cotton pests and showed that early planting reduces damage caused by thrips, cotton aphids, and boll weevils in the northern Texas Plains. Planting time was tested by Mitchell [31] to prevent insects damage on corn in Florida who showed that corn cobs in late planting, approximately two weeks after the ideal planting date, were severely damaged by earworm and FAW.

In the African context, the planting season depends on the effective rainfall [32]. However, in several countries in sub-Saharan Africa, farmers do not know how to plant at the ideal time. Several factors may explain this, including climatic variability expressed in terms of rainfall, input availability, weeds and pests, labor, etc. [33]. Alternatively, farmers may try to maximize crops with abundant rainfall during a cropping season by shifting planting times [34], which gives pests the opportunity to become well established [35]. Early planting means waiting for the effective onset of rains during the growing season to escape pest pressure [34]. This is when the plant benefits from the maximum amount of water and heat units. It grows rapidly and is more resistant to insect attack [36]. Niassy et al. [23] found that FAW infestations are usually low during periods with high rainfall. A late planting date does not often mean that the crop will be exposed to pests, as late planting is also a strategy to prevent the recurrence of certain pests that could affect the crop at the beginning of the season [30]. Rodríguez-del-Bosque et al. [37] found that FAW damage to maize cobs was highest in early planting, then decreased in mid-planting and increased further in late planting.

Since the invasion of the FAW in Africa, few studies have been conducted to assess the effect of planting date on the incidence of the pest. The studies by Nyabanga et al. [36] demonstrate that early planting reduces FAW infestations in maize crop in Zimbabwe, but Baudron et al. [38] did not find any effect of planting date on FAW infestations to maize in a farming survey in the same country. According to Baudron et al. [38], further research is needed to determine the effect of planting date on FAW outbreaks, which could be a cost-effective method of controlling the pest in African farmer context. Planting at the ideal moment is currently a challenge for most farmers in eastern DRC due to socioeconomic or environmental factors such as climate change. The timing of crop planting (planting date) has a significant impact on the sustainable management of FAW by smallholder farmers in South Kivu. The existence of multiple planting dates could lead to an overlap in the FAW cycle. The objectives of this study are to determine how different maize planting dates affect the FAW infestation, identify which planting periods are most susceptible to high FAW larval densities, in order to optimize maize planting schedules for pest management, and to investigate the distribution of various FAW larval stages in relation to maize planting periods and cropping system in South Kivu.

## Materials and methods

### Study sites

The study was conducted in Kabare territory in eastern DRC, located in the South Kivu province. This territory has an area of approximately 1.690 km$^2$ and its population, spread over two chiefdoms, Kabare and Nindja, is estimated at 535.114 inhabitants, with a density of 288 inhabitants per km$^2$ [39]. The altitude is between 1000 and 3250 m above sea level. The average annual precipitation and temperature are 1601 ± 154 mm and 19.67 ± 2.3˚C, respectively. Three sites were considered for investigation in this territory: Miti-Murhesa, Katana and Mudaka. These sites were selected based on their accessibility and are part of the corridor potentially suitable for FAW in South Kivu [40].

### Fields monitoring

Field monitoring was conducted in farmer fields of Kabare territory in March 2020 with a focus on the planting date and the degree of FAW infestation in the above-mentioned sites. The method consists of fields scouting of more than 100 fields and direct observation of plants, crops and surrounding areas. Information on the planting date and the level of FAW infestation was collected. It should be noted that two cropping seasons exist in South Kivu each year:

early season (season-1), which starts from September to January, and late season (season-2), from February to June. Based on information collected on the planting period and observations of farmers' fields infested by the FAW during this period (March-May 2020), a study was carried out during the cropping seasons from September to October 2020 and from February to March 2021.

Based on the differences in maize development stages from field to field and observations of the level of FAW infestation in the study area during the field monitoring period from March to May 2020, five planting dates separated by approximately two weeks were considered for each season. To identify the fields according to the planting dates, the transect method [41] was used in each selected site to track the fields. After identifying the first planting date for each season (01 September and 01 February), the remaining dates were identified each after approximately 2 weeks depending on the period considered. The geographical coordinates of the various fields were registered using a Global Positioning System (GPSMAP® 64s, GARMIN, United States) and allowed for the recognition of the fields during data collection of FAW infestation parameters. The dates of September 1, September 15, October 1, October 15 and October 30, 2020, were considered for season-1, while the dates of February 1, February 15, March 1, March 15 and March 30, 2021, were selected for season-2. For season-1, early planting comprised of September 1, September 15, October 1 and late planting included October 15, October 30. Whereas, for season-2 early planting comprised of February 1, February 15 and March 1 and late planting included March 15 and March 30.

Information on field characteristics was collected during field identification and survey and including field type (farmer or exploitation farm), cropping system (monoculture or intercropping), variety of planted maize, fertilization plan, and the surface area of each field (in square meters). Most of the fields planted after October 15 and March 15 were found in water-logged soils (usually marshlands). For the season-1, 45 fields were surveyed and distributed among the five planting dates. For the season-2, 45 new fields were selected based on the planting dates considered in that period. The choice of 45 fields was made to ensure an even distribution across the five planting dates. With nine fields allocated to each planting date, this allows for a balanced comparison and analysis between the different planting dates. This number ensures that sufficient data is collected for each date to produce reliable, statistically significant results. Overall, 90 fields were surveyed for the entire study period. The field allocation by planting date, site and season is presented in S1 Table.

## Assessment of fall armyworm infestation parameters

Three important parameters for assessing FAW infestation in the maize crop were considered: the percentage of plants infested by FAW, the damage severity determined using a rating scale updated by Toepfer et al. [42] and the larval density obtained by counting larvae. On the Davis scale, damage score of 4, 5 and 6 indicate the presence of several small, mid-sized to large, elongated lesions on whorl and furl leaves. A damage score of 7 and 8 indicates the presence of numerous elongated lesions of varying sizes on multiple whorl and furl leaves accompanied by several large holes with uniform to irregular shapes resulting from FAW feeding. Damage score 9 indicates that the whorled and rolled leaves are almost destroyed. To complete the three parameters, the type of FAW larval stage was determined in each field according to the planting dates. All the parameters were surveyed in each field when maize was at the V8 growth stage (8 leaves fully emerged) using the absolute (quadrat) count method [43]. Six quadrats shaped using stakes and ropes, each 20 m$^2$ in size, were randomly formed in each field using the W sampling method to collect FAW incidence. Magnifying glasses (PMS-054 of 6-fold increase) were used for close examination of plant damage and FAW larvae. To record

observations on plant damage, presence of larvae, clipboards and data sheets were used. Smartphones with Davis damage score photos were used to assess the severity of attack. The incidence and severity of FAW are high at the vegetative growth stage, which justifies the choice of the V8 stage for investigations. The stage corresponds to the 30th and 28th day after sowing for the season-1 and season-2, respectively.

To determine the larval stage of FAW, 50 larvae were randomly collected from maize plants in each field surveyed following Wyckhuys and O'Neil [44] methods. Early stages (L1 and L2) were thoroughly collected using a brush. Larvae were kept in rearing boxes (25cm × 17cm × 10cm) at a rate of 25 larvae per box and were fed with fresh maize leaves to avoid cannibalism in a rearing room of the Faculty of Agriculture and Environmental Sciences of the Université Evangélique en Afrique (UEA/Bukavu). Larvae from each field were soaked in 70% ethanol solution for approximately one minute on the same day of collection (3 hours after field investigation). The size of the larva in length was measured using a millimeter paper. A SOLO-MARK stereomicroscope—Science Lab 3D with an ocular micrometer was used to confirm insect identification and determine the width of the head capsule. The head capsule width and larval size (in length) values were compared to existing literature values [7,45] to determine the identity of the larval stages collected in each field.

## Statistical analysis

All the statistical analysis was performed on R version 4.1.3 [46]. The percentage of infested plants, the severity of the damages and the number of larvae were tested to compare the early and late planting group by student t-test for each planting season. These variables of both seasons are significantly and positively correlated with a correlation coefficient > 0.85 (p-value < 0.05). Therefore, the number of larvae as function of the independent explicative variables (i.e., fixed effects) was arbitrarily selected: the maize planting date (numerically converted in number of Julian day), the type of field, the parcel surface ($m^2$), the cropping system, the maize cultivar and the type of fertilizer. Given the unbalanced data gathering and the presence of pseudo-replication, generalized linear mixed-effects models (GLMMs) were performed using *lme4* R package [47]. The sampling sites were considered as factor effects (1|Sites). As counting data, Poisson distribution was selected to explain the distribution error. For the model selection, the second order Akaike Information Criterion (AICc) was assessed to classify the relative support given by the data to each model.

Redundancy analysis (RDA) was performed to assess (after removing colinear variable) the influence of latitude, maize planting date, incidence, type of field, parcel surface ($m^2$), cropping system, maize cultivar, and type of fertilizer on the larval stage composition from L1 to L6. All the explicative variables were previously standardized with *decostand* function from *vegan* R package [48]. The Hellinger's transformation was applied on the larval stage composition because it contains many zeros [49]. The model using *ordistep* function [48] (automatic stepwise model) was simplified by performing forward selection with 1000 permutations to select variables that are statistically important. Then, analysis of variance (ANOVA) was performed to analyze the significance of the RDA on the model and each selected variable of the model with 1000 permutations. All the graphics were generated with ggplot2 R package [50].

## Results

### Fall armyworm infestation varies with the planting period in South Kivu

In general, the number of larvae was great during the later planting dates in both seasons (October 15th, October 30th, March 15th and March 30th), and decreased at earlier planting dates (September 01st, September 15th, October 01st, February 01st, February 15th and March

## (a)

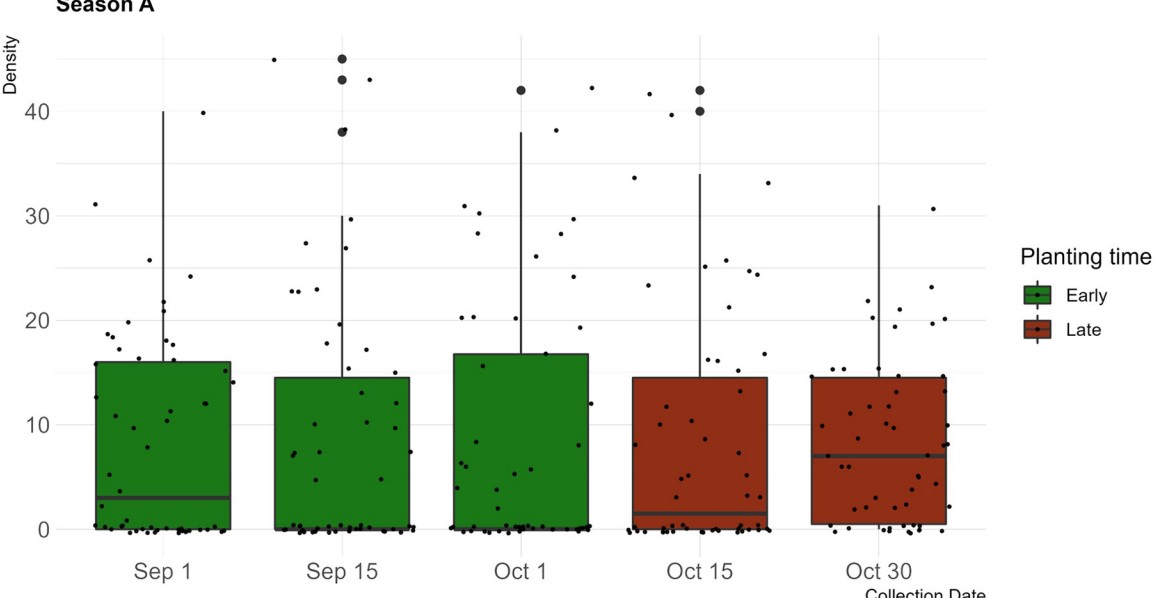

## (b)

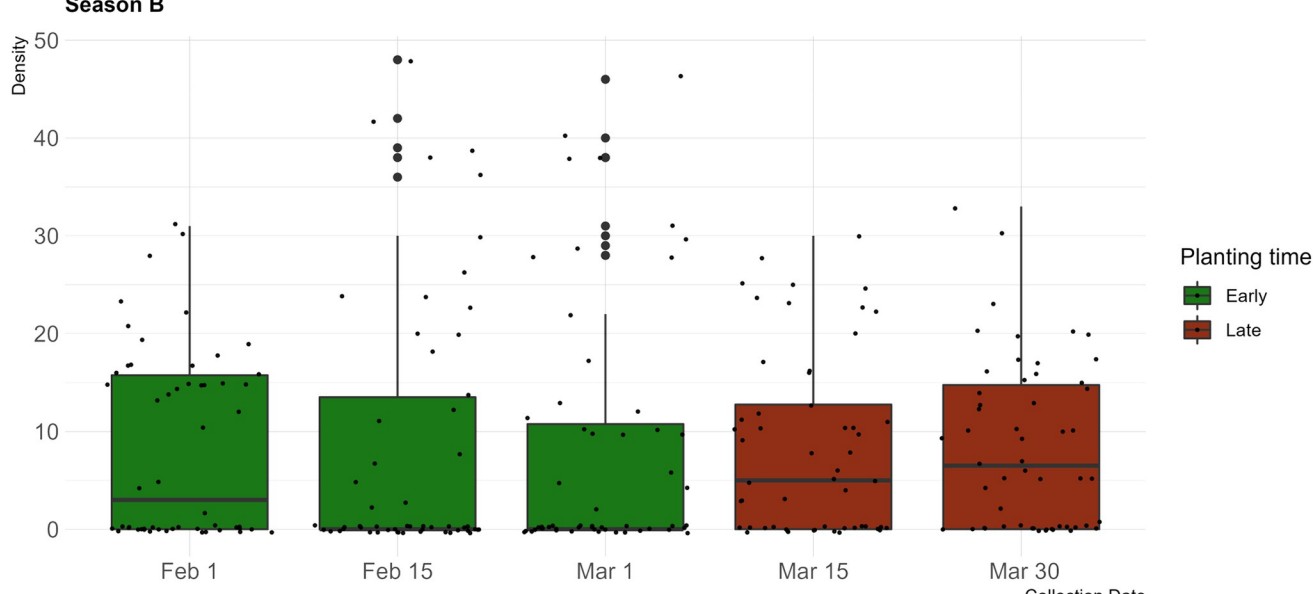

**Fig 1. Variation in larval density of *Spodoptera frugiperda* measured as the number of larvae per quadrat at different planting dates. (a):** Number of larvae for season-1; **(b):** Number of larvae for season-2.

01$^{st}$) as shown in Fig 1. In the context of season-1, the mean number of larvae reached 30.44 ± 6.90 in the late planting group against 17.78 ± 7.01 in the early planting group ($t_{Welch}$ = -6.38, df = 40.55, p-value < 0.001). A similar trend was observed in season-2, where late planting resulted in a higher larval population (33.27 ± 6.90) compared to early planting (19.00 ± 7.01) ($t_{Welch}$ = -6.75, df = 37.01, p-value < 0.001).

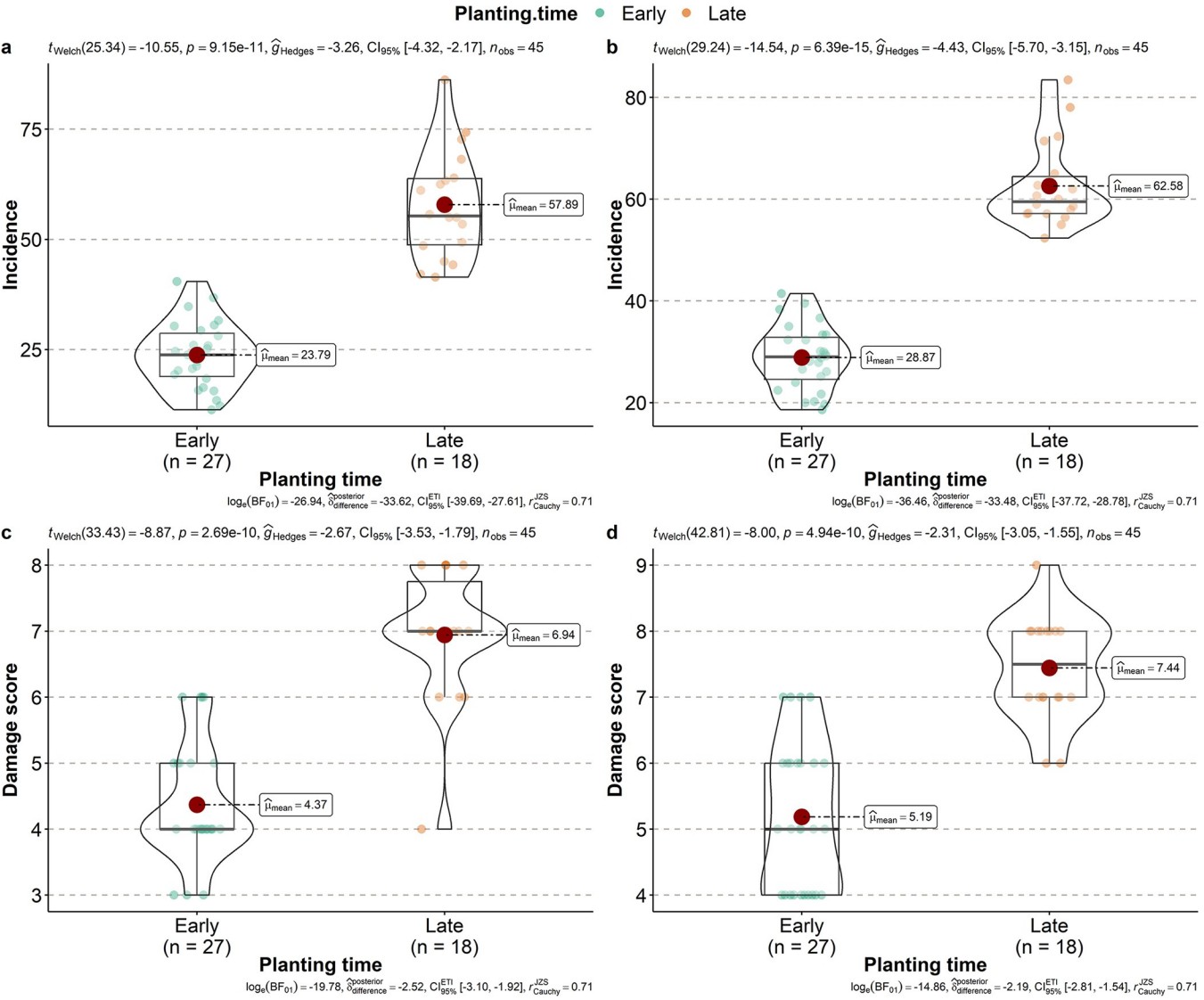

**Fig 2. Violin plot of incidence and severity of *Spodoptera frugiperda* in relation to planting time.** Overall statistical test with p-value and effect size with confidence intervals are shown on each plot. **a-b** represents the incidence for season-1 and season-2 respectively; **c-d** represents the severity for season-1 and season-2 respectively.

The incidence, which represents the proportion of plants with leaf damage by FAW, varied significantly based on the planting period in both season-1 and season-2 (Fig 2). The incidence reached its highest mean values in both seasons when planting was delayed, with rates of 57.89 ± 12.23% for early season ($t_{Welch}$ = -10.55, df = 25.34, p-value < 0.001) and 62.58 ± 8.41% for late season ($t_{Welch}$ = -14.54, df = 29.24, p-value < 0.001), as compared to early planting (23.79 ± 7.44% for season-1 and 28.86 ± 6.25% for season-2, respectively). This indicates an approximate 35% mean difference in incidence between late and early planting. In season-1, the mean damage score for late planting was 6.94 ± 0.99, whereas for early planting, it averaged at 4.37 ± 0.88 ($t_{Welch}$ = -8.87, df = 33.43, p-value < 0.001). Likewise, during season-2, a similar statistical pattern was observed, with mean values of 7.44 ± 0.78 for late planting and 5.19 ± 1.11 for early planting ($t_{Welch}$ = -7.99, df = 42.81, p-value < 0.001).

**Table 1. Summary of the results of the Generalized linear mixed models (GLMMs) fitted by maximum likelihood (Laplace approximation) for explaining the variability of the larval density of FAW with planting time.**

| Fixed effects | Season-1 | | | | |
|---|---|---|---|---|---|
| | Estimate | Std. Error | Z value | P value | AICc |
| **Intercept** | -1.21 | 0.43 | -2.76 | **0.005** | 282.00 |
| **Julian calendar** | 0.01 | 0.00 | 10.04 | **< 0.001** | |
| **Season-2** | | | | | |
| **Intercept** | 3.38 | 0.46 | 7.22 | **< 0.001** | 335.50 |
| **Type of field (Exploitation)** | -0.27 | 0.33 | -0.81 | 0.416 | |
| **Type of field (Farmer)** | -0.54 | 0.33 | -1.62 | 0.105 | |
| **Surface (m²)** | -0.04 | 0.06 | -0.68 | 0.4913 | |
| **Planting time (Late)** | 0.56 | 0.14 | 3.76 | **< 0.001** | |
| **Maize variety (M'Roma)** | 0.00 | 0.12 | 0.03 | 0.971 | |
| **Maize variety (SAM4 Vita)** | 0.13 | 0.19 | 0.69 | 0.484 | |
| **Maize variety (Z-M)** | -0.01 | 0.08 | -0.18 | 0.855 | |
| **Fertilizers (None)** | -0.06 | 0.16 | -0.40 | 0.686 | |
| **Fertilizers (NPK)** | -0.27 | 0.16 | -1.73 | 0.082 | |
| **Fertilizers (NPK+Manure)** | -0.40 | 0.23 | -1.70 | 0.088 | |
| **Fertilizers (Urea+Manure)** | -0.32 | 0.41 | -0.78 | 0.434 | |
| **Julian calendar** | 0.00 | 0.00 | 0.18 | 0.855 | |

## Population variation of FAW larvae

In both seasons, a total of five models were constructed, as detailed in S2 and S3 Tables. Using the Akaike Information Criterion corrected for small sample size (AICc) as a selection criterion, the Julian calendar model, presented as model 5 in Table 1, emerged as the most appropriate explanatory variable for elucidating the effect of planting period on larval density during the season-1. In contrast, for the season-2, the model 1 with all the explicative variables (also detailed in Table 1) was retained as the optimal model. Furthermore, it is noteworthy that, in the context of late season, the classification of planting periods, distinguishing between late and early planting, had a substantial and consistent influence on larval density across all study sites.

Larval density exhibits significant variation with planting period (Fig 3). Late planting is consistently correlated with increased larval density, indicating a robust association between late planting and increased FAW infestation, regardless of season. This association is statistically supported in both season-1 ($R^2$ = 0.670, p-value < 0.001) and season-2 ($R^2$ = 0.375, p-value < 0.001).

## Various larval stages in relation to planting time

Statistical differences were observed in density of each larval stages depending on the planting period as shown in Fig 4. In season-1, five larval stages were found in the same field at the V8 growth stage of maize. Only the L1 FAW larval stage was missing in the batch of collected larvae. In season-2, all larval stages were present in the same field. The tendency of results shows that later planting period has the highest density of each larval stage compared to early planting period. Considering the L1 larval stage, the density was recorded at late planting compared to early planting for season-2 (t = -4.20; df = 43; p-value < 0.001). In the case of L2 larval stage, the density was high for late planting compared to early planting at season-1 (t = -5.29, df = 43; p-value < 0.001) and season-2 (t = -3.73, df = 43; p-value < 0.001). Furthermore, for the L4 larval stages, the density was high for late planting compared to early planting at season-2 (t =

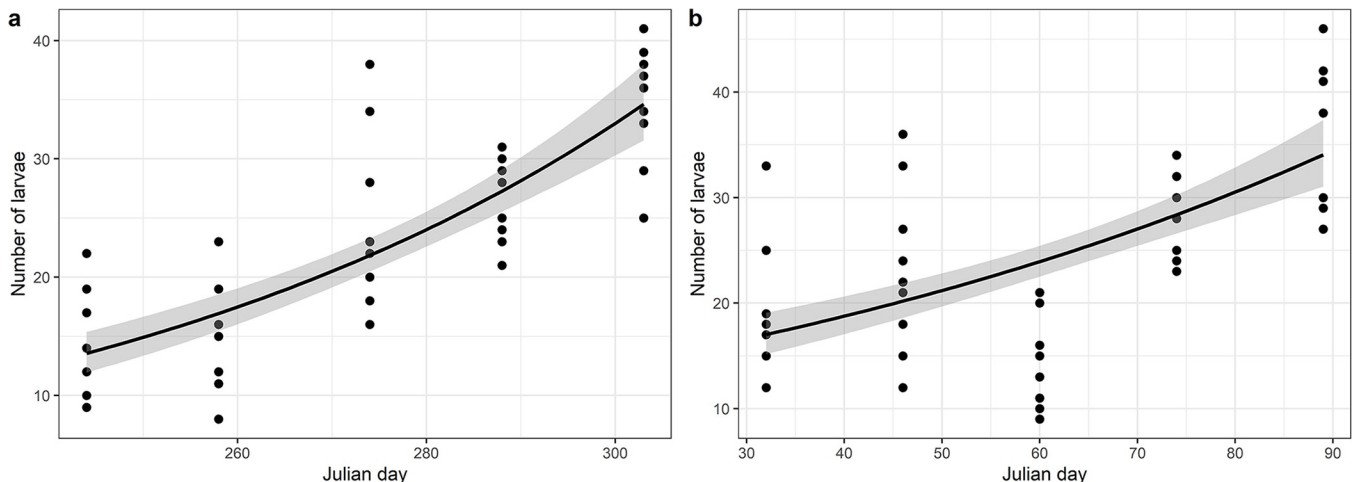

**Fig 3. Poisson prediction model of larval density with planting time.** Trend lines indicate model predictions, while dots represent observations. The grey area indicates the confidence interval set for the model at 95% level. **a** and **b** represent larval density prediction model for season-1 and season-2 respectively.

-3.44, df = 43, p-value < 0.01). No significant difference between early and late planting was observed for L3, L5, L6 at both seasons and L4 at season-1. These results indicate that, in addition to density, the presence of FAW is related to the category of larval stages found in the

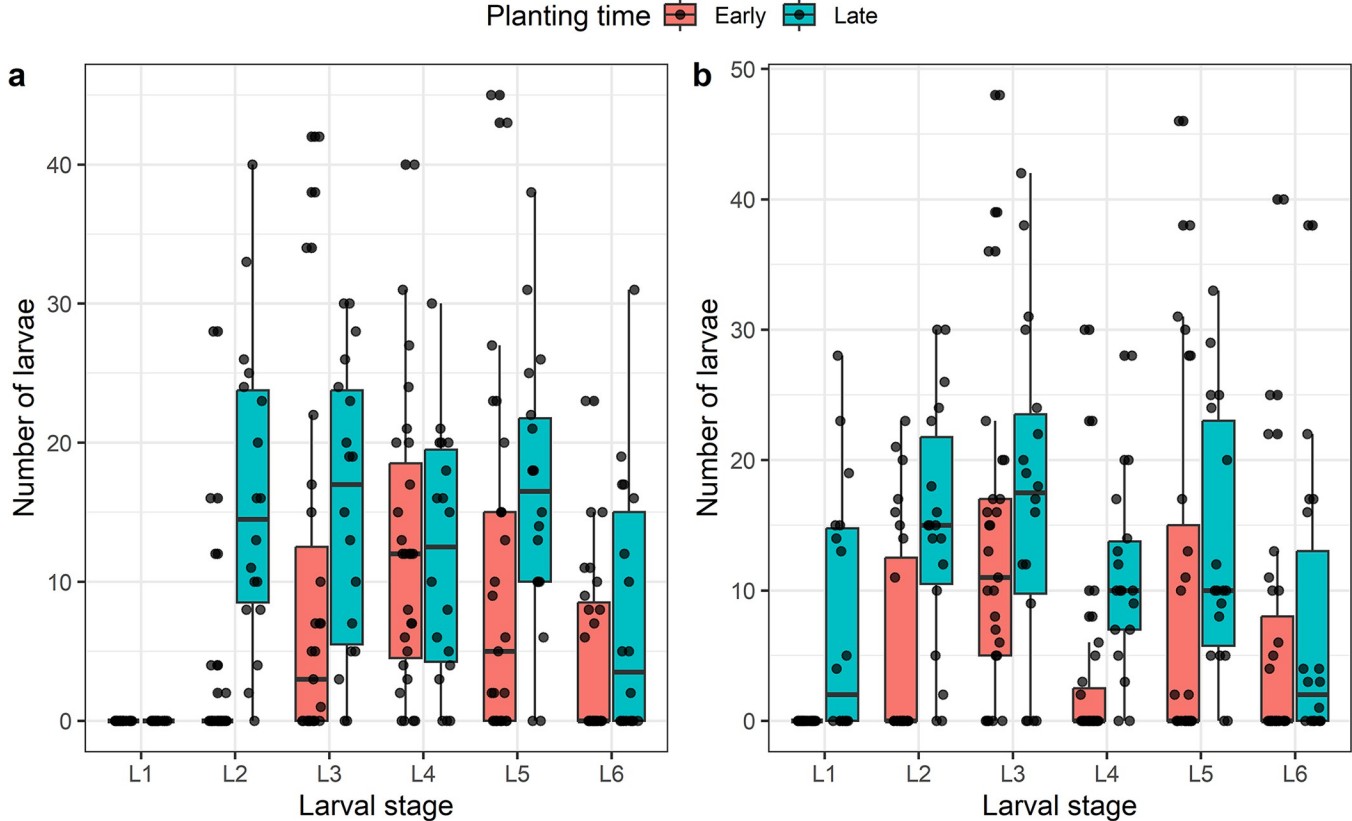

**Fig 4. Differences in larval density expressed as the number of individuals of each larval stage of *Spodoptera frugiperda* in relation to the planting time. a** and **b** represent larval stage density for season-1 and season-2 respectively.

**Table 2. Output analysis of variance (ANOVA) explaining the redundancy analysis (RDA) of *Spodoptera frugiperda* larval stage composition for season-1 and season-2.**

| | Season-1 | | | | | Season-2 | | | |
|---|---|---|---|---|---|---|---|---|---|
| **Variables** | **df** | **Variance** | **F** | **P value** | **Variables** | **df** | **Variance** | **F** | **P value** |
| **Type of field** | 2 | 0.22 | 1.62 | 0.149 | **Type of field** | 2 | 0.09 | 0.7 | 0.631 |
| **Cropping system** | 4 | 0.89 | 3.23 | 0.000 *** | **Cropping system** | 3 | 0.11 | 0.54 | 0.833 |
| **Maize variety** | 3 | 0.19 | 0.93 | 0.475 | **Maize variety** | 3 | 0.40 | 1.91 | 0.064 |
| **Fertilizers** | 5 | 0.33 | 0.97 | 0.500 | **Fertilizers** | 4 | 0.14 | 0.51 | 0.943 |
| **Julian calendar** | 1 | 0.41 | 6.03 | 0.002 ** | **Julian calendar** | 1 | 2.02 | 28.46 | 0.000 *** |
| **Incidence** | 1 | 0.07 | 1.12 | 0.312 | **Incidence** | 1 | 0.94 | 13.21 | 0.000 *** |
| **Latitude** | 1 | 0.26 | 3.78 | 0.025 * | **Longitude** | 1 | 0.05 | 0.75 | 0.499 |
| **Surface (m²)** | 1 | 0.04 | 0.67 | 0.607 | **Latitude** | 1 | 0.05 | 0.71 | 0.529 |
| **Residual** | 26 | 1.79 | | | **Surface (m²)** | 1 | 0.06 | 0.94 | 0.408 |
| | | | | | **Residual** | 27 | 1.92 | | |

Significance codes:

*** p < 0.001

** p < 0.01

* p < 0.05.

same field, regardless of the planting period. Consequently, the species is more frequent throughout the growing season.

The summary results of the RDA analysis of variance (Table 2) show the variables that had the most significant influence on the composition of larval stage of FAW in both seasons.

The projection fields of the three sites considered in this study on the main planes formed by RDA1 and RDA2 do not show any differences between the sites in the two seasons (Fig 5). In season-1, three variables including cropping system, planting date (numerically expressed as Julian calendar) and latitude influenced the larval stage at the three sites considered. High densities of L2 and L3 larvae are much more associated with late planting in early season. Considering the cropping system, maize monoculture, maize intercropping with cassava and maize intercropping with bean systems had a significantly greater influence on the presence of FAW L4, L5 and L6 larvae, whereas maize intercropping with groundnut and maize intercropping with soybean systems seemed to influence FAW L2 and L3 larvae. L1 larvae of FAW were found in all cropping systems. In season-2, two variables had an influence on the larval stage composition of FAW. These were the planting date and incidence. High densities of L4, L5 and L6 larvae are much more associated with late planting. The highest incidence occurs when L4, L5 and L6 larvae are present, typically associated with late planting, whereas the incidence is low when L1, L2, L3 larvae are found in early sown fields.

## Discussion

The fall armyworm is already well established in eastern DRC [40]. Its damage to maize crops varies according to season and agroecological zones [43,51]. In general, considering the phenological stages of maize, FAW attacks start once the first leaves unfold, precisely at early whorls [16] depending on the planting period. The fall armyworm has a rapid and dynamic life cycle that is influenced by environmental conditions, especially temperature [7]. Its development from egg to adult can occur in about 30–40 days in warm climates. On the other hand, maize, depending on its variety and growing conditions, progresses through distinct stages—germination, vegetative stages (V), and reproductive stages (R). Results from this study show

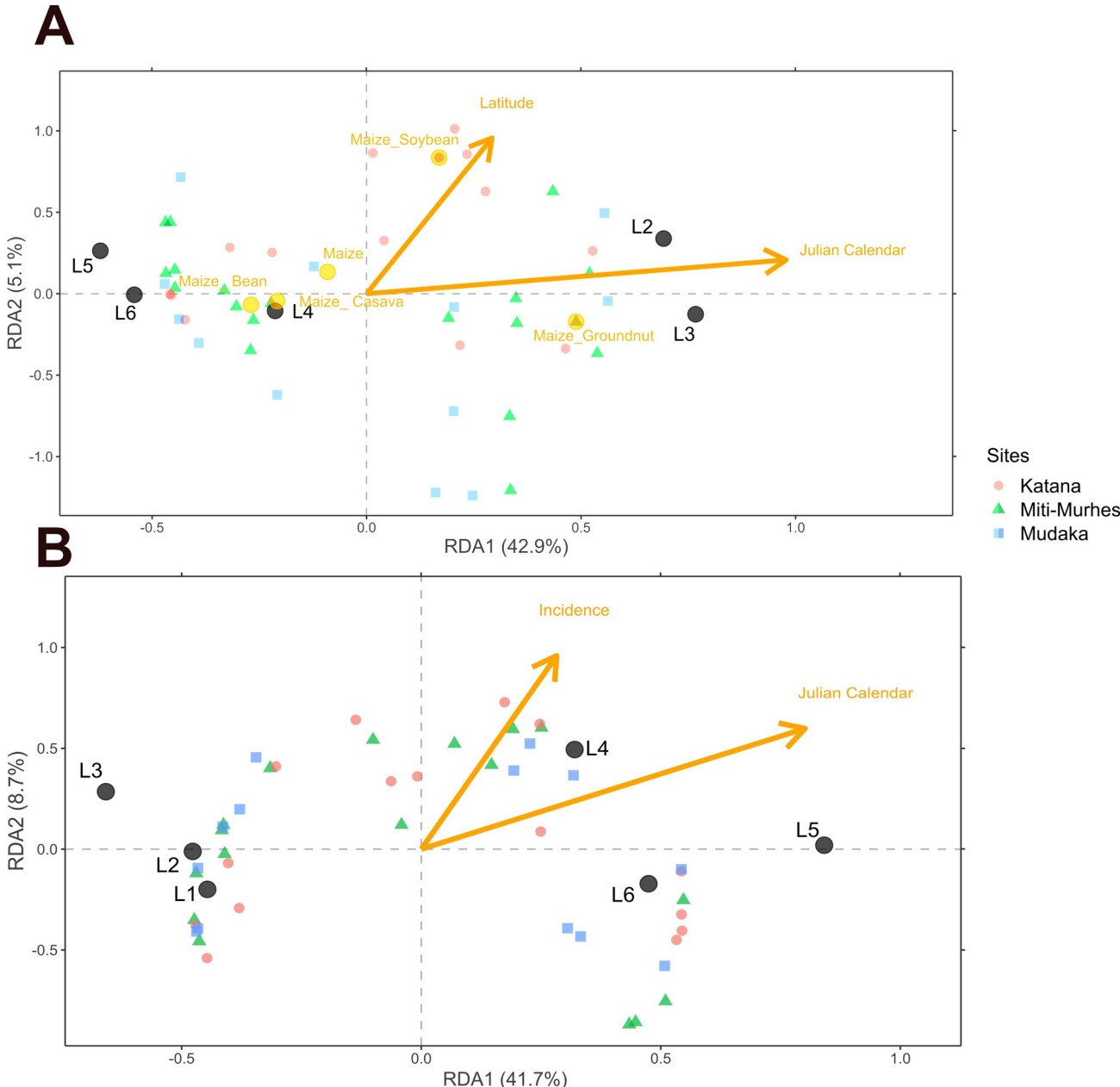

**Fig 5.** Redundancy analysis triplot of larval stage composition for season-1 (A) and season-2 (B). Sites scores are grouped by collection sites: Red dot for Katana; green triangle for Miti-Murhesa; blue square for Mudaka. Black dots represent the larval stage of *Spodoptera frugiperda*. Orange solid line vectors represent significant quantitative environmental variables. Orange dots represent significant centroid of qualitative environmental variable only for season-1.

that late-planted fields were much more severely infested by FAW than early-planted fields. The populations of pests in the early or late-planted fields resulted from a temporal separation of pest and crop [52]. When maize is planted late in the season, its vegetative and early reproductive stages, which are critical for yield formation, may coincide with the peak population of FAW. Incidence and severity had the highest mean values in both seasons when planting was delayed compared to early planting. The mean difference in incidence between late and early planting is approximately 35%. Results from Nyabanga et al. [36] showed that planting date

had a significant effect on both FAW incidence and severity, with higher values for late planting. In contrast, Baudron et al. [38] found no effect of planting date on FAW infestation. However, both authors conducted their research in the same country. Even within the same country, environmental conditions such as temperature, humidity, and rainfall can vary significantly between regions or even within a short distance, and FAW Infestations have been documented to vary across regions within the same country [53]. These microclimatic differences can affect the development and behavior of the FAW and maize growth patterns, potentially influencing the pest's impact based on planting dates. The study of Nyabanga et al. [36] may have been conducted in an area more sensitive to these conditions, where late planting created a more favorable environment for FAW. The exact periods of the studies may differ slightly, leading to variations in weather conditions during early or late planting. A specific year or location could experience unusual weather patterns (e.g., a prolonged rainy season), which may influence FAW populations differently. FAW is a polyphagous pest, but maize is one of its preferred hosts [5]. When maize is planted late, there may be fewer alternative host plants available for the pest, particularly in regions with a monocropping system. This reduced availability of other food sources may drive FAW to concentrate on the remaining late-planted maize fields, increasing the intensity of infestation. In the absence of a continuous supply of host plants to attack, FAW may be present in some areas at different times [23].

In this study, FAW larval density was higher in late than in early plantations. According to Nyabanga et al. [36], early planted crops escape pest pressure because the phenology of the crop does not coincide with the period of pest abundance. Hruska and Gould's [15] results showed that early maize growth stages are more tolerant to lepidopteran attack than later stages. It is known that maize yield is not always affected when FAW infestation occurs at the vegetative growth stage [54], as the plant is able to compensate for damage when in optimal soil and climatic conditions [6,22]. In regions where multiple maize crops are grown in a season, early-planted maize can harbor low populations of FAW, which gradually increase as the season progresses. Late-planted maize becomes an attractive target for these escalating FAW populations, which are now larger and more aggressive. This occurs because early maize crops allow FAW to complete their first or second generation, producing higher numbers of adults that will lay eggs on the late-planted crops [55]. Thus, delayed planting inadvertently exposes maize to higher pest pressure due to the compounding population growth of FAW across generations. Late planting often occurs when environmental conditions such as temperature and moisture are more favorable for FAW reproduction and survival. Warmer temperatures can accelerate FAW development, leading to more rapid population increases [56]. In general, early planting is linked to effective rainfall. In South Kivu, FAW infestation is less severe during the season-1, a season characterized mainly by heavy rainfall and low temperatures, conditions that are unfavorable for FAW [43]. According to Niassy et al. [23], rainfall affects the dynamics of FAW, but the impact of this parameter on FAW populations in Africa has not been fully investigated. Nboyine et al. [51] found a correlation between rainfall and FAW moth capture, suggesting that rainfall and relative humidity contribute positively to moth abundance. The timing of rainfall can dictate the most favorable planting dates for avoiding FAW infestations. For instance, planting too early during the rainy season may lead to high pest pressure as the crop coincides with peak FAW moth activity. Conversely, if planting is delayed until after heavy rainfall, there might be reduced infestation levels due to natural washing away of larvae or egg mortality [20]. Furthermore, early planting after optimal rainfall allows the maize crop to be in optimal condition by efficiently using water and heat units early in the growing season [6,34]. Late-planted maize is typically exposed to climatic conditions that might not favor optimal growth, making it more vulnerable to stress from both pests and environmental factors, thus compounding the impact of FAW infestations. In a study by

Rodríguez-del-Bosque et al. [37], FAW damage was highest at the earliest planting dates, decreased at intermediate dates and increased at the latest dates. Considering that the FAW is a polyphagous species with multiple generations that can be observed from 4 to 6 per year, depending on optimal climatic conditions expressed in degree-days [7,57], it is obvious that late plantations will have more attacks during the seasons. In Africa, studies show that the number of generations varies according to the seasons and climatic conditions throughout the year [23], compared to conditions in the Americas, where the species migrates when conditions are no longer optimal [9]. In addition, late planting may disrupt the natural biological control mechanisms that exist earlier in the season when natural enemy populations are higher [55]. As a result, FAW populations in late-planted maize may experience less predation or parasitism, allowing them to proliferate more freely. Late planting is not always disadvantageous for the crop in terms of pests, as Slosser [30] found that delaying planting predicted the infestation of certain pests, in this case boll weevil, and did not systematically increase the pest problem. Farmers in the Kabare area and other regions facing FAW infestations must contend with the pest's rapid reproduction, wide host range, and climatic influences that affect its lifecycle. By aligning farming practices, particularly planting dates, with an understanding of local climate dynamics and pest behavior, farmers can significantly reduce the risk of FAW damage. Farmers can enhance their ability to time planting effectively by utilizing climate forecasting tools and local weather monitoring [58]. Seasonal forecasts for rainfall and temperature patterns can provide valuable guidance on when to plant and what to expect in terms of pest pressure. In regions like Kabare, where maize is a major crop, synchronizing planting dates at the community level can help reduce FAW pressure across an entire area. When crops are at similar growth stages, it becomes easier to target FAW at a community level using integrated pest management (IPM) strategies such as biological controls, chemical sprays, or pheromone traps [6,59].

Looking at the developmental cycle of FAW, the presence of larvae, regardless of stage, should be uniform in the same field, with small variations depending on the feeding ability of each larva [57]. However, in some situations there may be differences in size due to delayed oviposition and female longevity [7,45]. According to the results of this study, five larval stages of FAW were found in the same maize field in the season-1 and all larval stages in the season-2 at V8 maize stage, regardless of the planting period. This result is surprising because under normal conditions, the first generation of FAW that emerges at the V3 stage can complete its development from early larval stages (L1) to adult, mate and re-infest the maize crop at the reproductive stage during the same cropping season [16]. This is generally the case in eastern Congo, where temperatures easily reach 25°C, ideal for the development of FAW [43]. In many regions where FAW is already endemic, multiple overlapping generations can be observed on the same maize plant [16]. Behaviorally, when population densities of FAW in a field are high, females lay eggs indiscriminately on all maize plants [8]. At this point, differences in larval size can be observed. The indiscriminate egg-laying behavior observed in females may be due to their desire to give the larvae at least some chance of development, given their highly adaptable, almost omnivorous nature [7]. This behavior may allow the larvae to eventually find a suitable host plant for further growth. In this study, we did not trap FAW moths to understand the results related to the presence of different larval stages in the same field. However, studies by Nboyine et al. [51] show that there is a positive correlation between the trapping of adults and the abundance of larvae.

The trend in the results shows that the late planting period has the highest density of each larval stage compared to the early planting period. High densities of L2 and L3 larvae are much more associated with late planting in season-1, while high densities of L4, L5 and L6 larva, more voracious [7,57], are much more associated with late planting in season-2. The presence of these larval stages in large numbers during the season-2 explains why late planting during

this period is dangerous, not only in the Kabare area where the study was conducted, but also throughout the Great Lakes sub-region [43]. The results of this study show that the incidence is highest when L4, L5 and L6 larvae are present at the V8 stage, often associated with late planting, and decreases when L1, L2, L3 larvae are present in early-sown fields. This is contrary to the results of Cokola et al. [43], who found that the presence of young larvae, generally L1, L2 and L3, cause numerous lesions resulting in high incidence. The maize monoculture, maize-cassava intercropping and maize-bean intercropping systems had a significantly greater influence on the presence of FAW L4, L5 and L6 larvae, whereas the maize-groundnut and maize-soybean intercropping systems appeared to have an influence on FAW L2 and L3 juvenile larvae. Understanding the relationship between cropping systems and pests is crucial for sustainable agricultural production. Crop diversification influence pest dynamics in general [25] and FAW specifically [6,24]. Maize-legume intercropping has been studied as an alternative FAW management method in two different models. The first model is a conventional maize-legume system (soybean, bean, groundnut,...) [26,60] and the second is a push-pull system [24,61]. Maize-legume intercropping improves soil health while promoting plant vigor, especially through nitrogen fixation, which improves local atmospheric conditions at the plot level [62]. In addition, intercropping limits larval movement between plants and prevents females from laying eggs on maize by emitting semiochemicals [25,61]. The abundance, diversity and activity of natural enemy arthropods also increase in this system, helping to reduce pest populations [6,25]. In a study by Udayakumar et al. [26], maize intercropping with faba bean, *Desmodium* sp. and groundnut recorded significantly higher rates of egg parasitism and FAW predation. The juvenile larval stages (L1, L2, L3) found in intercropping systems in this study are the ones most likely to be parasitized by insects, according to Durocher-Granger et al. [63] results, which explains the low incidence associated with their presence in maize intercropped with soybean and groundnut. Considering the push-pull system, results from Sobhy et al. [27] showed that companion crop volatiles repel FAW, while attracting its natural enemy parasitoids, explaining why the system has fewer larvae and lower infestations than monoculture maize.

## Conclusions

The study demonstrates that FAW infestation in South Kivu is significantly influenced by planting time. Late planting, generally 30 October in season-1 and 30 March in season-2, consistently leads to higher larval densities and greater damage to maize crops compared to early planting (15 September in season-1 and 01 Mars in season-2). This trend is observed across both seasons, with late planting also associated with a higher incidence of damage and the presence of advanced larval stages. The variation in larval stages and their density further emphasizes the importance of planting periods in managing FAW infestations, indicating that early planting may reduce the risk of severe FAW outbreaks. In addition, the presence of maize crops during the dry season in the marshlands, as was the case in this study, further complicates the situation. Knowing the ideal planting time in South Kivu is challenging because there are practically no weather stations or forecasting systems that can establish a direct relation between FAW infestation rates and climatic variables such as rainfall and temperature. The existence of weather stations and forecasting systems would enable farmers to choose the ideal planting time to effectively manage FAW and maximize maize production in the region.

## Supporting information

**S1 Table. Allocation of the number of fields being monitored according to planting dates, seasons and study locations.**
(DOCX)

**S2 Table. Summary of the results of the selection of Generalized linear mixed models (GLMMs) to explain the variability of larval density with other variables in early season.** (DOCX)

**S3 Table. Summary of the results of the selection of Generalized linear mixed models (GLMMs) to explain the variability of larval density with other variables in late season.** (DOCX)

## Acknowledgments

We would like to thank the smallholder farmers for facilitating the identification of planting dates and access to fields for surveys at their respective locations. To all the students and colleagues who participated in the data collection.

## Author Contributions

**Conceptualization:** Marcellin Cuma Cokola, Rudy Caparros Megido, Frédéric Francis.

**Data curation:** Marcellin Cuma Cokola, Yannick Mugumaarhahama.

**Formal analysis:** Grégoire Noël.

**Funding acquisition:** Espoir B. Bisimwa, Frédéric Francis.

**Investigation:** Marcellin Cuma Cokola, Yannick Mugumaarhahama.

**Methodology:** Marcellin Cuma Cokola, Grégoire Noël, Yannick Mugumaarhahama.

**Supervision:** Espoir B. Bisimwa, Frédéric Francis.

**Validation:** Frédéric Francis.

**Writing – original draft:** Marcellin Cuma Cokola, Rudy Caparros Megido.

**Writing – review & editing:** Grégoire Noël, Yannick Mugumaarhahama, Rudy Caparros Megido, Espoir B. Bisimwa, Frédéric Francis.

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
