## [Decision Letter · Decision Letter 0]

23 Jul 2024

PONE-D-24-16438Planting date in South Kivu, eastern DR Congo: a real challenge for the sustainable management of Spodoptera frugiperda (Lepidoptera: Noctuidae) by smallholder farmersPLOS ONE

Dear Dr. Cokola,

Thank you for submitting your manuscript to PLOS ONE. After careful consideration, we feel that it has merit but does not fully meet PLOS ONE’s publication criteria as it currently stands. Therefore, we invite you to submit a revised version of the manuscript that addresses the points raised during the review process.

We look forward to receiving your revised manuscript.

Kind regards,

Allah Bakhsh

Academic Editor

PLOS ONE

Journal Requirements:

"Université Evangélique en Afrique with funds from Pain pour Le Monde (Project A-COD-2023-0035)"

"We would like to thank the smallholder farmers for facilitating the identification of planting dates and access to fields for surveys at their respective locations. To all the students and colleagues who participated in the data collection. This work was supported with funds from Pain Pour Le Monde (Brot für die Welt) to Université Evangélique en Afrique (Project A-COD-2023-0035)."

"Université Evangélique en Afrique with funds from Pain pour Le Monde (Project A-COD-2023-0035)"

"The authors declare that there is no conflict of interest"

5. We note that [Figure 1] in your submission contain [map/satellite] images which may be copyrighted. All PLOS content is published under the Creative Commons Attribution License (CC BY 4.0), which means that the manuscript, images, and Supporting Information files will be freely available online, and any third party is permitted to access, download, copy, distribute, and use these materials in any way, even commercially, with proper attribution. For these reasons, we cannot publish previously copyrighted maps or satellite images created using proprietary data, such as Google software (Google Maps, Street View, and Earth). For more information, see our copyright guidelines: http://journals.plos.org/plosone/s/licenses-and-copyright.

6. We are unable to open your Supporting Information file [Supporting Information - Compressed/ZIP File Archive]. Please kindly revise as necessary and re-upload.

Reviewers' comments:

Reviewer's Responses to Questions

**Comments to the Author**

1. Is the manuscript technically sound, and do the data support the conclusions?

Reviewer #1: Yes

Reviewer #2: Yes

2. Has the statistical analysis been performed appropriately and rigorously? 

Reviewer #1: Yes

Reviewer #2: Yes

3. Have the authors made all data underlying the findings in their manuscript fully available?

Reviewer #1: Yes

Reviewer #2: Yes

4. Is the manuscript presented in an intelligible fashion and written in standard English?

Reviewer #1: Yes

Reviewer #2: Yes

5. Review Comments to the Author

Reviewer #1: The manuscript titled " Planting date in South Kivu, eastern DR Congo: a real challenge for the sustainable management of Spodoptera frugiperda (Lepidoptera: Noctuidae) by smallholder farmers’’ explains the impact of the planting dates of maize on fall army worm infestation that will be helpful to devise management strategies for this pest. However, numerous areas of the documents need to be improved before it is considered fit for publication. It can be accepted for publication after major revision.

I have few comments/suggestions that I hope the authors will consider. All can be addressed with some serious re-writing.

L 23-24: Rewrite as ‘’planting date of maize and FAW infestation’’.

L 31: How is it possible that only 5 larval stages present in the field. Give explanation.

L 37-39: Give main conclusion of your study.

L 49: Do not start the sentence with abbreviation.

L 61: Rewrite as ‘’Democratic Republic of Congo (DRC)’’.

L 101: No need to write Democratic Republic of Congo.

L 115-121: Describe method of monitoring briefly. It is better to write as early season (season-1) and late season (season-2) to remove further confusion throughout the manuscript.

L 133-136: Modify the whole sentence. Clearly write planting dates for each season separately viz. For season-1, early planting comprised of September 1, September 15, October 1 and late planting included October 15, October 30. Whereas, for season-2 early planting comprised of February 1, February 15 and March 1 and late planting included March 15 and March 30.

L 149: Briefly describe rating scale.

L 157: What was the method to collect 1st instar? The difference in result may be related to collection of different stage. As 1st instar are considerably smaller in size and vulnerable. Explain this point.

Q: How many cropping systems were included in this study? To justify sentence in L 255-259 mention cropping systems in materials and methods section.

Q: Clear picture of meteorological data of three sites is lacking. It is important to show corelation of FAW incidence with weather parameters.

Q: Statistical analysis must explain why you selected these models.

L 199: Replace late season with season-2. Replace early and late season with season-1 and season-2 respectively, in result as well as discussion portion.

L 236: Why L1 stage was missing? How many times larval batches were collected in season-1 or early season? Not clear.

L 328-329: What does it mean? ‘’As the sampling was random, the probability of not finding a stage was not negligible, especially if the larva is small’’. It looks sampling error; how will you justify this?

L 356-357: Mention names of parasites. Did you find parasitized larvae from field collected population of FAW?

L 363-375: Is it conclusion or summary.

Several aspects of this study are lacking that are mentioned below.

• What is main hypothesis of this study?

• What is the conclusion of this study?

• Which planting time is best?

• What is novelty of this work?

• Only surveys were conducted to record population of FAW. While, to define appropriate planting time a thorough field study via planting date experiments is required.

• The abstract does not convey the relevance of the findings.

• Explain the practical aspects of inter cropping in the introduction. The introduction does not provide sufficient background on FAW and their impact on maize crops. However, the presentation of the study's aims and scope should be better organized and clearer.

• Although the materials and methods section provide thorough information on data gathering methods and survey procedures, but writeup is much confusing.

• The discussion explains the findings, but it falls short of comparing them to earlier investigations. It would be more appropriate to rewrite discussion.

• The authors should edit and clarify the abstract, introduction, materials and methods, findings, and discussion sections to improve the manuscript's publication readiness. They should also extensively check the manuscript for errors and inconsistencies. Furthermore, fixing any missing information and maintaining uniformity in formatting and citation style would improve the manuscript's overall quality.

• Please provide a more accurate conclusion to your study.

Finally, for the most part, the paper is well organized. In addition, more attention should be given to details (e.g., omission of articles, word usage). More care – More thought.

Reviewer #2: Manuscript No : PONE-D-24-16438

Introduction: The introduction is well-crafted, providing a thorough and compelling rationale for the study. It sets a solid foundation for the subsequent sections of the paper.

Suggestions for Improvement

1. Ensure consistent citation style throughout the article. For instance, consider using the same format for author names (e.g., “J. E. Smith, 1797” or “Smith, 1797”).

2. Clarify Contradictions: When discussing contradictory findings from previous studies (e.g., lines 64 and 65, Overton et al. vs. Harrison et al.), briefly explain potential reasons for these discrepancies.

3. Lines 212-217 can go to the material and methods section to describe the

Materials and methods: The methodology section is robust and well-documented, providing a clear framework for conducting the study on FAW infestation dynamics in relation to planting dates in South Kivu.

Suggestions for Improvement

1. Justification of Sampling: Provide a brief justification for why 45 fields were chosen per season. This could include power analysis considerations or practical constraints

2. Clearly describe the method and tools used to collect data on the incidence of the FAW on the field.

Discussion: The discussion effectively analyses the impact of planting periods on FAW infestation in maize crops, supported by rigorous statistical analysis and relevant literature. Addressing the suggested areas for improvement would enhance the clarity and depth of your manuscript, making it more impactful for readers interested in agricultural pest management strategies

Suggestions for Improvement:

1. Causality vs. Correlation: While the discussion outlines the correlation between late planting and increased FAW infestation, further exploration into potential causal mechanisms would enrich the discussion. For example, discussing how delayed planting affects FAW life cycle synchronization with maize growth stages could provide deeper insights.

2. Influence of Climate Factors: it was mentioned briefly, rainfall and temperature as factors influencing FAW dynamics, but a more detailed discussion on how these climatic variables specifically interact with planting dates and FAW infestation could enhance the robustness of your conclusions. Exploring how climatic conditions impact FAW moth activity and larval development would be insightful.

3. Practical Implications: Consider expanding on the practical implications of the findings for maize farmers in the Kabare area and similar regions. Discussing strategies for optimizing planting dates to minimize FAW damage could provide actionable insights for agricultural practitioners.

4. Comparison with Contrasting Studies: You mentioned studies with contrasting findings regarding the impact of planting dates on FAW infestation. Discussing potential reasons for these discrepancies and how they relate to your study's findings would strengthen the discussion section.

6. PLOS authors have the option to publish the peer review history of their article (what does this mean?). If published, this will include your full peer review and any attached files.

Reviewer #1: No

Reviewer #2: **Yes: **Ken Okwae Fening

---

## [Author Response · Author response to Decision Letter 0]

4 Oct 2024

RE: Response to reviewer comments

September 26th, 2024

Dear Editor,

Thank you for giving us the opportunity to submit a revised version of our manuscript entitled “Planting date in South Kivu, eastern DR Congo: a real challenge for the sustainable management of Spodoptera frugiperda (Lepidoptera: Noctuidae) by smallholder farmers” to PLoS ONE. We appreciate you and the reviewers for your precious time in reviewing our paper and providing valuable comments. We have studied comments carefully and have made correction in line with the suggestions made by you and the reviewers. We apologize for the delay in the review. The revision interfered with my thesis defense activities. We thank the reviewers for their insightful comments that improve the manuscript. We hope the manuscript after careful revisions meet your high standards.

Here is a point-by-point response to the journal requirements, reviewers’ comments and concerns. 

Journal Requirements

Authors’ Response: Thanks for the corrections, the manuscript has been formatted accordingly. Changes have been made wherever necessary.

"Université Evangélique en Afrique with funds from Pain pour Le Monde (Project A-COD-2023-0035)"

Authors’ Response: Thank you.The text has been added to the funder statement.

"We would like to thank the smallholder farmers for facilitating the identification of planting dates and access to fields for surveys at their respective locations. To all the students and colleagues who participated in the data collection. This work was supported with funds from Pain Pour Le Monde (Brot für die Welt) to Université Evangélique en Afrique (Project A-COD-2023-0035)."

"Université Evangélique en Afrique with funds from Pain pour Le Monde (Project A-COD-2023-0035)"

Authors’ Response: Thank you. The text was removed from the manuscript.

"The authors declare that there is no conflict of interest"

Authors’ Response: Thank you so much. Please change in the online submission.

5. We note that [Figure 1] in your submission contain [map/satellite] images which may be copyrighted. All PLOS content is published under the Creative Commons Attribution License (CC BY 4.0), which means that the manuscript, images, and Supporting Information files will be freely available online, and any third party is permitted to access, download, copy, distribute, and use these materials in any way, even commercially, with proper attribution. For these reasons, we cannot publish previously copyrighted maps or satellite images created using proprietary data, such as Google software (Google Maps, Street View, and Earth). For more information, see our copyright guidelines: http://journals.plos.org/plosone/s/licenses-and-copyright.

We require you to either (1) present written permission from the copyright holder to publish these figures specifically under the CC BY 4.0 license, or (2) remove the figures from your submission

Authors’ Response: Thank you for your comment. The figure was removed from the manuscript.

6. We are unable to open your Supporting Information file [Supporting Information - Compressed/ZIP File Archive]. Please kindly revise as necessary and re-upload.

Authors’ Response: Thank you for the observation. The files have been uploaded individually.

Reviewer 1 

Comment 1: L 23-24: Rewrite as ‘’planting date of maize and FAW infestation’’.

Authors’ Response: Thank you. Corrected in the manuscript (L24).

Comment 2: L 31: How is it possible that only 5 larval stages present in the field. Give explanation. 

Authors’ Response: Thank you for the observation. It was found that the larvae varied in size when they were collected according to planting date. This led us to understand why there were several larval stages in the same field. All stages were found in one field. This is unusual according to the life cycle of the pest. Its means that the moths are abundant in the region and lay their eggs at different times during the season.

Comment 3: L 37-39: Give main conclusion of your study. 

Authors’ Response: Thank you for the suggestion. The conclusion was added (L37-39).

Comment 4: L 49: Do not start the sentence with abbreviation.

Authors’ Response: Thank you for the suggestion. Corrected (L49).

Comment 5: L 61: Rewrite as ‘’Democratic Republic of Congo (DRC)’’.

Authors’ Response: Thank you for the suggestion. Corrected (L64).

Comment 6: L 101: No need to write Democratic Republic of Congo.

Authors’ Response: Thank you. Corrected (L113).

Comment 7: L 115-121: Describe method of monitoring briefly. It is better to write as early season (season-1) and late season (season-2) to remove further confusion throughout the manuscript.

Authors’ Response: Thank you so much. The method was described briefly (L135-138), and the correction was made where necessary regarding the designation of the seasons.

Comment 8: L 133-136: Modify the whole sentence. Clearly write planting dates for each season separately viz. For season-1, early planting comprised of September 1, September 15, October 1 and late planting included October 15, October 30. Whereas, for season-2 early planting comprised of February 1, February 15 and March 1 and late planting included March 15 and March 30.

 Authors’ Response: Thank you so much for the suggestions. Correction was made as suggested (L155-158).

Comment 9: L 149: Briefly describe rating scale.

Authors’ Response: Thank you for the suggestions. The rating scale was described in the manuscript (L174-181).

Comment 10: L 157: What was the method to collect 1st instar? The difference in result may be related to collection of different stage. As 1st instar are considerably smaller in size and vulnerable. Explain this point.

Authors’ Response: Thank you for the observation. Small larvae (L1 and L2) were thoroughly collected using a brush. This sentence has been added to the text for clarification (L189-190).

Comment 11: Q: How many cropping systems were included in this study? To justify sentence in L 255-259 mention cropping systems in materials and methods section.

Authors’ Response: Thank you for the question. Two cropping systems were considered: monoculture and combining maize with other crops. This is indicated in the methodology (L162-165).

Comment 12: Q: Clear picture of meteorological data of three sites is lacking. It is important to show correlation of FAW incidence with weather parameters. 

Authors’ Response: Thank you for your pertinent comment. Initially, the aim was to correlate infestation data and climatic parameters according to planting dates at each site. Unfortunately, the limited number and almost non-existence of weather stations made this impossible. Subsequently, a revision of the agricultural calendar is being planned, which will allow the installation of weather stations to provide real time climatic data.

Comment 13: Q: Statistical analysis must explain why you selected these models.

Authors’ Response: Thank you for the question. This is specified in the results (L263-264).

Comment 14: L 199: Replace late season with season-2. Replace early and late season with season-1 and season-2 respectively, in result as well as discussion portion.

Authors’ Response: Thank you for the suggestions. Changes have been made throughout the manuscript.

Comment 15: L 236: Why L1 stage was missing? How many times larval batches were collected in season-1 or early season? Not clear.

Authors’ Response: Thank you for the comment. For each planting date, larvae were collected only once at the 8-leaf stage. Collection was random in each field. The aim was not to target a particular stage, but to collect a number of larvae of unknown stage. The stage was determined in the laboratory a few hours after collection.

Comment 16: L 328-329: What does it mean? ‘’As the sampling was random, the probability of not finding a stage was not negligible, especially if the larva is small’’. It looks sampling error; how will you justify this?

Authors’ Response: Thank you for the comment. As I said previously, the sampling was random. The sentence has been deleted to avoid any misunderstanding (L460-463).

Comment 17: L 356-357: Mention names of parasites. Did you find parasitized larvae from field collected population of FAW?

Authors’ Response: Thank you for the comment. The study did not include FAW parasites. Another study is currently underway to address this aspect of the research. Parasitoids such as Coccigydium luteum, Chelonus bifoveolatus, Diadegma sp., Charops diversipes, Drino quadrizonula, etc. have so far been collected including entomopathogenic fungi (Metarhizium and Beauveria) in the South Kivu farming system.

Comment 18: L 363-375: Is it conclusion or summary. 

Author’s Response: Thank you for the observation. The text was corrected, and a conclusion was proposed (L496-508).

Several aspects of this study are lacking that are mentioned below.

• What is main hypothesis of this study?

Author’s Response: Thank you for the question. Hypothesis was added (L114-116).

• What is the conclusion of this study?

Author’s Response: Thank you for the question. The conclusion was added (L496-508).

• Which planting time is best?

Author’s Response: Thank you for the question. This information was added in the conclusion (L447-454) and in the abstract (L30-31).

• What is novelty of this work? 

Author’s Response: Thank you for the question. Firstly, studies on planting dates and FAW invasion are scarce. In this study, we found that planting date was a key factor in understanding FAW invasion in South Kivu, with late planting exacerbating FAW damage. Secondly, the presence of all larval stages in the same field allows FAW to occur frequently throughout the growing season. Finally, the presence of these larval stages varies according to the cropping system used by farmers.

• Only surveys were conducted to record population of FAW. While, to define appropriate planting time a thorough field study via planting date experiments is required. 

Author’s Response: Thank you for the observation. Several planting dates were identified before the dates used in this study were considered. There was no need for experimentation as the farmers' fields already showed differences in maize growth. The study was based on the availability of fields according to the selected dates (See Appendix).

• The abstract does not convey the relevance of the findings. 

Author’s Response: Thank you for the observation. The abstract has been improved.

• Explain the practical aspects of inter cropping in the introduction. The introduction does not provide sufficient background on FAW and their impact on maize crops. However, the presentation of the study's aims and scope should be better organized and clearer.

 Author’s Response: Thank you. Suggestions have been added in the introduction (L60-62; 79-85). The presentation of objectives has been better organized and clarified (L116-120).

• Although the materials and methods section provide thorough information on data gathering methods and survey procedures, but writeup is much confusing. 

Author’s Response: Thank you for the observation. The text of the materials and methods section has been improved based on your suggestions.

• The discussion explains the findings, but it falls short of comparing them to earlier investigations. It would be more appropriate to rewrite discussion. 

Author’s Response: Thank you for the proposition. The discussion section was improved as suggested also by the second reviewer.

• The authors should edit and clarify the abstract, introduction, materials and methods, findings, and discussion sections to improve the manuscript's publication readiness. They should also extensively check the manuscript for errors and inconsistencies. Furthermore, fixing any missing information and maintaining uniformity in formatting and citation style would improve the manuscript's overall quality.

Author’s Response: Thank you for the observations. The correction was made for all the above-mentioned aspects. 

• Please provide a more accurate conclusion to your study. 

Author’s Response: Thank you for the proposition. Already done in the conclusion section and in the abstract.

Finally, for the most part, the paper is well organized. In addition, more attention should be given to details (e.g., omission of articles, word usage). More care – More thought.

Authors’ Response: Thank you.

Reviewer 2 

Introduction: The introduction is well-crafted, providing a thorough and compelling rationale for the study. It sets a solid foundation for the subsequent sections of the paper.

Authors’ Response: Thank you so much for your appreciation.

Suggestions for Improvement

1. Ensure consistent citation style throughout the article. For instance, consider using the same format for author names (e.g., “J. E. Smith, 1797” or “Smith, 1797”).

Authors’ Response: Thank you for the observation. In this case, the author is the one who first described the species. I remove the year to conform to the journal's standards and to avoid it being considered a citation in the text (L46).

2. Clarify Contradictions: When discussing contradictory findings from previous studies (e.g., lines 64 and 65, Overton et al. vs. Harrison et al.), briefly explain potential reasons for these discrepancies.

Authors’ Response: Thank you for the comment. The reasons were added in the manuscript (L68-71).

3. Lines 212-217 can go to the material and methods section to describe the damage severity (Highlighted text is well suited for the materials and methods section of manuscript).

Authors’ Response: Thank you for the suggestion. The text has been moved to the methodology (L174-181).

Materials and methods: The methodology section is robust and well-documented, providing a clear framework for conducting the study on FAW infestation dynamics in relation to planting dates in South Kivu.

Authors’ Response: Thank you.

Suggestions for Improvement

1. Justification of Sampling: Provide a brief justification for why 45 fields were chosen per season. This could include power analysis considerations or practical constraints

Authors’ Response: Thank you for the comment. We provide in the methodology a brief justification (L168-172).

2. Clearly describe the method and tools used to collect data on the incidence of the FAW on the field.

Authors’ Response: Thank you for the comment. The methods and tools used to collect data on the incidence of the FAW were provided (L189-193).

Discussion: The discussion effectively analyses the impact of planting periods on FAW infestation in maize crops, supported by rigorous statistical analysis and relevant literature. Addressing the suggested areas for improvement would enhance the clarity and depth of your manuscript, making it more impactful for readers interes

---

## [Decision Letter · Decision Letter 1]

14 Nov 2024

Planting date in South Kivu, eastern DR Congo: a real challenge for the sustainable management of Spodoptera frugiperda (Lepidoptera: Noctuidae) by smallholder farmers

PONE-D-24-16438R1

Dear Dr. Cokola,

We’re pleased to inform you that your manuscript has been judged scientifically suitable for publication and will be formally accepted for publication once it meets all outstanding technical requirements.

Kind regards,

Allah Bakhsh

Academic Editor

PLOS ONE

Additional Editor Comments (optional):

Reviewers' comments:

Reviewer's Responses to Questions

**Comments to the Author**

1. If the authors have adequately addressed your comments raised in a previous round of review and you feel that this manuscript is now acceptable for publication, you may indicate that here to bypass the “Comments to the Author” section, enter your conflict of interest statement in the “Confidential to Editor” section, and submit your "Accept" recommendation.

Reviewer #1: All comments have been addressed

2. Is the manuscript technically sound, and do the data support the conclusions?

Reviewer #1: Yes

3. Has the statistical analysis been performed appropriately and rigorously? 

Reviewer #1: Yes

4. Have the authors made all data underlying the findings in their manuscript fully available?

Reviewer #1: Yes

5. Is the manuscript presented in an intelligible fashion and written in standard English?

Reviewer #1: Yes

6. Review Comments to the Author

Reviewer #1: Dear Author,

I am pleased to see that you have addressed the previous comments thoroughly, which has greatly improved the manuscript’s clarity and overall quality.

I am satisfied with the responses and revisions and believe the manuscript is now suitable for publication, pending any additional editorial review.

7. PLOS authors have the option to publish the peer review history of their article (what does this mean?). If published, this will include your full peer review and any attached files.

Reviewer #1: No

---

## [Editor Report · Acceptance letter]

18 Nov 2024

PONE-D-24-16438R1 

PLOS ONE

Dear Dr. Cokola, 

I'm pleased to inform you that your manuscript has been deemed suitable for publication in PLOS ONE. Congratulations! Your manuscript is now being handed over to our production team.

Kind regards, 

on behalf of

Dr. Allah Bakhsh 

Academic Editor

PLOS ONE